# Relationship between youth cardiometabolic health and physical activity in medical records

Bethany Forseth[1,2]*, Janelle R. Noel-MacDonnell[3,4,5], Sarah Hampl[2,4], Jordan A. Carlson[2,4], Kelsee Halpin[5,6], Ann Davis[2,7], Tarin Phillips[8], Robin P. Shook[2,4]

1 Department of Physical Therapy, Rehabilitation Science, & Athletic Training, University of Kansas Medical Center, Kansas City, KS, United States of America, 2 Center for Children's Healthy Lifestyles & Nutrition, Kansas City, MO, United States of America, 3 Department of Health Services and Outcomes Research, Children's Mercy Kansas City, Kansas City, MO, United States of America, 4 Department of Pediatrics, Children's Mercy Kansas City, Kansas City, MO, United States of America, 5 University of Missouri—Kansas City School of Medicine, Kansas City, MO, United States of America, 6 Division of Pediatric Endocrinology and Diabetes, Children's Mercy Kansas City, Kansas City, MO, United States of America, 7 Department of Pediatrics, University of Kansas Medical Center, Kansas City, KS, United States of America, 8 Kirksville College of Osteopathic Medicine, A.T. Still University, Kirksville, MO, United States of America

* bhanson4@kumc.edu

**Data Availability Statement:** Data cannot be shared publicly because of it is identifiable patient data from electronic medical records. Data may be requested from the Children's Mercy Hospital

## Abstract

### Background

Thers is limited research examining modifiable cardiometabolic risk factors with a single-item health behavior question obtained during a clinic visit. Such information could support clinicians in identifying patients at risk for adverse cardiometabolic health. We investigated if children meeting physical activity or screen time recommendations, collected during clinic visits, have better cardiometabolic health than children not meeting recommendations. We hypothesized that children meeting either recommendation would have fewer cardiometabolic risk factors.

### Methods and findings

This cross-sectional study used data from electronic medical records (EMRs) between January 1, 2013 through December 30, 2017 from children (2–18 years) with a well child visits and data for $\geq 1$ cardiometabolic risk factor (i.e., systolic and diastolic blood pressure, glycated hemoglobin, alanine transaminase, high-density and low-density lipoprotein, total cholesterol, and/or triglycerides). Physical activity and screen time were patient/caregiver-reported. Analyses included EMRs from 63,676 well child visits by 30,698 unique patients (49.3% female; 41.7% Black, 31.5% Hispanic). Models that included data from all visits indicated children meeting physical activity recommendations had reduced risk for abnormal blood pressure (odds ratio [OR] = 0.91, 95%CI 0.86, 0.97; p = 0.002), glycated hemoglobin (OR = 0.83, 95%CI 0.75, 0.91; $p = 0.00006$), alanine transaminase (OR = 0.85, 95%CI 0.79, 0.92; $p = 0.00001$), high-density lipoprotein (OR = 0.88, 95%CI 0.82, 0.95; $p = 0.0009$), and triglyceride values (OR = 0.89, 95%CI 0.83, 0.96; $p = 0.002$). Meeting screen time recommendations was not associated with abnormal cardiometabolic risk factors.

Institutional Data Access / Ethics Committee (contact bhanson4@kumc.edu for more details and guidance) for researchers who meet the criteria for access to confidential data. Resubmission update: Thank you for supporting transparency with data and publicly available data. Our dataset was created from electronic medical records from one hospital (the hospital is identified by name in the paper on the ethics approval line) and contains many data aspects that may compromise patient privacy. We did reach out to our IRB to identify best options for sharing the data. The IRB responded that our study was not approved to share patient data. We are able to share deidentified data with other researchers who complete a data user sharing agreement with Children's Mercy; this agreement will need to be completed with the Children's Mercy Office of Research Business Partners. As corresponding author, if another researchers requests the data I am happy to help with working through this process the data user agreement with the Children's Mercy Institution.

**Funding:** The author(s) received no specific funding for this work.

**Competing interests:** "The authors have read the journal's policy and have the following competing interests: BF received salary support from the National Institutes of Health for research not directly related to this project (F32DK128982). This does not alter our adherence to PLOS ONE policies on sharing data and materials."

## Conclusion

Collecting information on reported adherence to meeting physical activity recommendations can provide clinicians with additional information to identify patients with a higher risk of adverse cardiometabolic health.

## Introduction

Engaging in physical activity and reducing sedentary behaviors (eg, screen time) improves cardiometabolic health and weight status in children [1, 2]. The prevalence of cardiometabolic diseases, including obesity, type 2 diabetes, and hypertension are rising among youth in the United States [3–7]. Due to the overwhelming research supporting the benefits of engaging in healthy lifestyle behaviors, it is recommended that children (6–17 years) engage in at least 60 minutes of moderate-to-vigorous physical activity per day [8, 9]. Additionally, new recommendations on screen time and social media encourage parents to develop personalized media plans for their family and emphasize screen time breaks or limiting screen time to promote health, sleep, and physical activity, among other benefits [10].

Obtaining information on whether or not an individual meets physical activity and screen time recommendations may provide clinicians additional indications of patients who are at a higher risk of adverse cardiometabolic health. Screening for risk factors allows an opportunity to identify and talk to the patient/family about specific modifiable behavioral factors that may be contributing to health problems. Additionally, early detection of adverse cardiometabolic health can help with prevention. Some health care systems have integrated questions on physical activity and screen time behaviors into electronic medical records (EMR), allowing for this information to be easily obtained and viewed during routine visits [11–13]. Research on one-to-two item physical activity questions indicate they are valid for measuring exercise and they can detect differences between groups based on sex and weight status [13–15]. Previous research in children indicates that responses to physical activity and screen time questions are associated with weight status and systolic blood pressure [15–17]. Further, counseling on specific behavioral risk factors is recommended in individuals with certain cardiometabolic risks (eg, higher sugar sweetened beverage intake and its association with elevated alanine transaminase and risk for nonalcoholic fatty liver disease) [18]. There is a need for a broader and more comprehensive look at prevalent and modifiable cardiometabolic risk factors and the ability of lifestyle questions within an EMR to identify pediatric patients at risk for adverse cardiometabolic health.

Therefore, the purpose of this study was to examine the relationship between responses to healthy lifestyle questions within an EMR and modifiable cardiometabolic risk factors in youth. The primary objective of this study was to investigate if children meeting physical activity or screen time recommendations, obtained and recorded in EMRs, have fewer modifiable cardiometabolic risks than children not meeting the recommendations. The second objective is to examine differences in modifiable cardiometabolic risk factors in youth grouped by weight and activity status (ie, under/normal weight meeting physical activity recommendations, under/normal weight not meeting physical activity recommendations, overweight/obese meeting physical activity recommendations, overweight/obese not meeting physical activity recommendations). Results from these objectives are important because they will support clinicians, who are already using EMR systems with these questions, to better identify patients

who may be at a higher risk for adverse cardiometabolic health outcomes and provide guidance on behavior modification [19, 20].

## Methods

### Overview

This cross-sectional study reviewed EMRs data from January 1, 2013, through December 30, 2017. EMR data were obtained from a large midwestern pediatric primary care clinic based in a tertiary care children's hospital. Study sites included four primary care clinics within the hospital system. The local Institutional Review Board approved the study and all procedures (IRB #16050373).

**Participants and data selection.** On September 30, 2020 data on patient visits for youth between the ages of 2–18 years were accessed for research purposes. The data included identifiable information. For each patient, we used all clinic visits during this time period that met the following criteria: complete responses were collected for the physical activity and screen time survey questions, height and weight were collected, and at least one of seven cardiometabolic risk indicators of interest was captured within 90 days after the routine visit. Participants (n = 43) were excluded if their EMR included positive labs for type 1 diabetes anti-bodies (ie, GAD, Islet antigen 2 antibody, Insulin Ab, or Zinc Transporter) at any point in time.

In instances where there was more than one day of lab values recorded within the 90-day window, the most complete data was selected. If more than one blood pressure value for the visit was recorded in the EMR, researchers used the latest (second) value obtained during the visit. In instances where there was more than one well child visit within 90 days prior to the lab visit (n = 106), the well child visit closest to the lab date was selected. Data were deidentified after merging the files containing the well child visit and the lab visit.

### Measures

**Participant demographics and clinical characteristics.** Child demographics including sex (male, female), race/ethnicity (white, Black, Hispanic/Latino, other race not listed), age category (2–5 years, 6–11 years, 12–18 years) and insurance type (commercial, government, no insurance, unknown) were obtained from EMR. Additionally, maternal and paternal comorbidities were pulled from the EMR, including endocrine (Type 2 diabetes mellitus, pre-diabetes, gestational diabetes, obesity), and cardiovascular-related conditions (hypertension, heart disease, gestational hypertension, hyperlipidemia, and dyslipidemia). Days between the well child visit and lab data were calculated (range 0–90 days).

**Lifestyle assessment.** Questions regarding a child's amount of physical activity and screen time on a typical day were asked during routine visits for children at least 2 years of age. The physical activity question asked "On a typical day, how many minutes does your child spend in active play/exercise (breathing harder or sweating);" response options were: none, <15, 15, 30, 45, 60, or ≥90 minutes. The screen time question asked "On a typical day, how many hours is your child in front of a screen (TV, computer, video game, cell phone);" responses included: none, 1, 1.5, 2, 2.5, 3, 3.5, 4, 4.5, ≥5 hours. For the purposes of this study, responses were dichotomized into meeting or not meeting recommendations. Meeting physical activity recommendations included responses of 60 minutes or more of activity and meeting screen time recommendations included responses of 2 hours or less of screen time, following previous numerical recommendations [8, 21, 22].

**Cardiometabolic risk factors.** Data on systolic and diastolic blood pressure and six cardiometabolic labs were obtained from the EMR. These labs included glycated hemoglobin (A1c), alanine transaminase (ALT), blood pressure, high-density lipoprotein (HDL), low-density

lipoprotein (LDL), total cholesterol, and triglycerides. Within the EMR, LDL values were calculated (LDL = total cholesterol–HDL–(triglycerides/5)). For more accurate triglyceride assessment, patients were asked to arrive fasting, though there was no method of confirming fasting status. Lab values were dichotomized into 'normal' or 'abnormal' (eg, at a higher risk or diagnostic value of disease) based on expert consensus or established clinical guidelines [18, 21, 23, 24] (S1 Table). Individual cardiometabolic labs that were absent from a panel of labs were considered 'missing' for some analyses, whereas for other analyses they were considered 'normal' lab values, as detailed below. Researchers decided this based on recommendations by the American Academy of Pediatrics and Bright Futures which outline periodicity and indications for recommended lab (eg, universal dyslipidemia screening at age 9–11 years) [10, 25].

**Body mass index percentile.** Height and weight measures obtained during well child visits were pulled from the EMR. Body mass index (BMI) percentile was calculated according to each child's BMI according to their sex and age at the visit based on the Centers for Disease Control and Prevention (CDC) criteria [24]. BMI percentiles were classified as underweight (<5th percentile), normal weight (5–84.9th percentile), overweight (85th– 94.9th percentile), or obese (≥95th percentile) [26].

## Data cleaning

Prior to merging the well child visit, blood pressure and lab data, any labs taken outside of the hospital system, or without numerical data were removed. However, the visit itself was not removed because other labs may have been conducted in the clinic on that day. Data from other clinic visits which included blood pressure and lab values, within 90 days after the well child visit, were merged via Python 3.8.3, pandas 1.0.5, numpy 1.18.5.

Once data were merged, extreme or biologically implausible values were removed from the dataset. For lab values, extreme values were removed using guidelines from the Pediatric Obesity Weight Evaluation Registry (POWER) which are based on expert consensus or established clinical guidelines (S1 Table) [27]. Extreme or biologically implausible height, weight and BMI values were removed using the CDC guidelines [28].

**Statistical analyses.** Participant-level demographics and visit-level information are described through frequencies and percentages. Data was examined using two samples of patients: 1) only eligible patients with measured data for a given cardiometabolic health indicator (ie, Lab Sample); and 2) all eligible patients, with missing data for a given cardiometabolic health indicator coded as a 'normal' value (ie, Full Sample). The same statistical modeling approaches were used for the two samples, which involved binary logistic regressions to examine the associations of meeting physical activity and screen time recommendations, and BMI category (independent variables), with clinically abnormal blood pressure or lab values (dependent variables). Generalized linear mixed models were used to account for the nesting of visits within patients. Fixed coefficients included sex, age grouping, race, insurance, parent comorbidity, and days between visits. Patient identification number was allowed to vary randomly to capture individual differences. The physical activity and screen time variables were entered together into the same model and each dependent variable was tested in a separate model. Covariates included sex, age, race, insurance type, mother comorbidity, father comorbidity, and days between routine visit and blood pressure or lab values. A second set of models added BMI category X physical activity groupings as an additional independent variable, defined as under/normal weight and meeting physical activity recommendations, under/normal weight and not meeting physical activity recommendations, overweight/obese and meeting physical activity recommendations, overweight/obese and not meeting physical activity recommendations. Odds ratios for abnormal lab values and their corresponding 95%

confidence intervals were calculated. Post-hoc analyses used the Bonferroni adjustment to assess statistical significance per aim (significant at $\alpha = 0.0036$). All statistical analyses were run using SPSS (version 27).

## Results

Once data were merged and cleaned, the final sample included 63,676 visits from 30,698 unique children (Fig 1). Patients in the dataset had between 1 to 7 well child visits; most patients only had one visit (41.1%) followed by two visits (27.9%) or three visits (18.5%) within the observed time period. Table 1 displays participant-level demographic characteristics and visit-level information. At the visit level, a majority of the children (85.9%) met physical

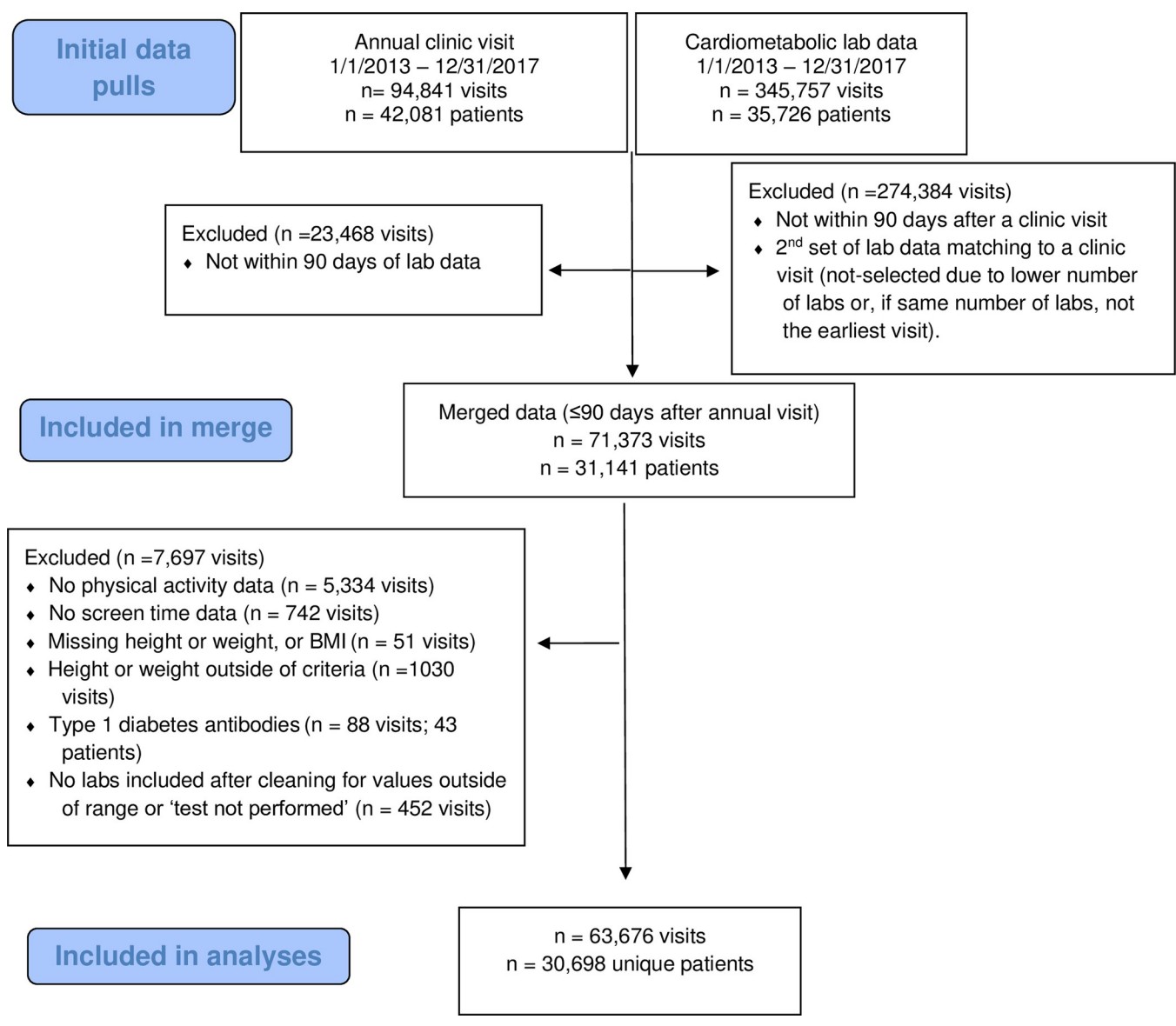

**Fig 1. Consort flow diagram for data included in analyses.**

**Table 1. Participant and visit demographics and characteristics.**

| Characteristic | No. (%) |
|---|---|
| **Participant Level** | **N = 30,698 participants** |
| Female | 15,128 (49.3%) |
| Race / Ethnicity | |
| Black/ African American | 12,804 (41.7%) |
| Hispanic/Latino | 9,672 (31.5%) |
| Other race not listed | 3,250 (10.6%) |
| White | 4,972 (16.2%) |
| Insurance | |
| Commercial / Private | 4,339 (14.1%) |
| Government / Public | 23,668 (77.1%) |
| No insurance | 2187 (7.1%) |
| Unknown | 504 (1.6%) |
| Family history of disease | |
| Mother comorbidity | 1,372 (4.5%) |
| Father comorbidity | 2,023 (6.1%) |
| **Visit Level** | **N = 63,676 visits** |
| Age | |
| 2–5 years | 22,178 (34.8%) |
| 6–11 years | 28,859 (45.3%) |
| 12–18 years | 12,630 (19.8%) |
| BMI Percentile | |
| ≤5th percentile | 1,799 (2.8%) |
| 5th-85th percentile | 38,189 (60.0%) |
| 85th–<95th percentile | 10,745 (16.9%) |
| 95th—<120% of the 95th percentile | 9,277 (14.6%) |
| 120%—<140% of the 95th percentile | 2,850 (4.5%) |
| ≥140% of the 95th percentile | 816 (1.3%) |
| Meeting recommendations | |
| PA recommendations | 54,723 (85.9%) |
| Screen time recommendations | 39,130 (61.5%) |

No.: number/frequency of participants.

activity recommendations and screen time recommendations (61.5%); with half of the sample (54.6%) reportedly meeting both recommendations. Table 2 displays the frequency of visits with blood pressure and lab data.

**Table 2. Frequency of abnormal cardiometabolic risk factors.**

| | A1c | ALT | BP | HDL | LDL | TC | TG |
|---|---|---|---|---|---|---|---|
| No. of visits with data for risk factor of interest | 8,936 | 9,165 | 62,729 | 13,649 | 13,512 | 13,661 | 13,624 |
| No. of abnormal results | 2,027 | 5,374 | 14,100 | 4,367 | 277 | 596 | 4,815 |
| % of abnormal values from visits with lab and BP data | 22.7% | 58.6% | 22.5% | 32.0% | 2.1% | 4.4% | 35.3% |
| % of abnormal lab/BP from all visits[a] | 3.1% | 8.4% | 22.1% | 6.9% | 0.4% | 0.9% | 7.6% |

[a] If visit did not contain value for blood pressure or a specific cardiometabolic lab the value was considered to be within a normal/healthy range (N = 63,676); Visits with no blood pressure or lab panel (ie, missing data for all labs of interest and blood pressure) were excluded; No.- number; A1c- glycated hemoglobin; ALT—Alanine transaminase; BP–blood pressure; HDL–high density lipoprotein; LDL–low density lipoprotein; TC–total cholesterol; TG—triglycerides

**Table 3. Odds ratios of meeting physical activity and screen time recommendations and under/normal weight body mass index classification with abnormal cardiometabolic risk indicators.**

| | A1c | ALT | BP | HDL | LDL | Total cholesterol | Triglycerides |
|---|---|---|---|---|---|---|---|
| **Lab sample** | | | | | | | |
| Sample size | 8,936 | 9,165 | 62,729 | 13,649 | 13,512 | 13,661 | 13,624 |
| Meeting physical activity recommendation[a] | 0.81 (0.71, 0.94) | 1.06 (0.95, 1.19) | **0.91 (0.85, 0.96)*** | 0.95 (0.86, 1.05) | 0.99 (0.80, 1.23) | 0.91 (0.76, 1.09) | 0.97 (0.87, 1.07) |
| Meeting screen time recommendation[b] | 0.97 (0.86, 1.09) | 1.02 (0.93, 1.12) | 0.98 (0.94, 1.02) | 1.0 (0.92, 1.09) | 0.95 (0.80, 1.14) | 0.94 (0.80, 1.09) | 1.06 (0.97, 1.15) |
| under/normal classification[c] | 0.69 (0.53, 0.90) | **0.45 (0.39, 0.53)*** | **0.60 (0.57, 0.62)*** | **0.34 (0.31, 0.38)*** | 0.78 (0.64, 0.95) | **0.76 (0.64, 0.90)*** | **0.25 (0.23, 0.28)*** |
| **Full sample[d]** | | | | | | | |
| Meeting physical activity recommendation[a] | **0.83 (0.75, 0.91)*** | **0.85 (0.79, 0.92)*** | **0.91 (0.86, 0.97)*** | **0.88 (0.82, 0.95)*** | 0.98 (0.86, 1.12) | 0.94 (0.83, 1.07) | **0.89 (0.83, 0.96)*** |
| Meeting screen time recommendation[b] | 0.95 (0.86,1.03) | 0.95 (0.89, 1.00) | 0.98 (0.93,1.02) | 0.93 (0.88, 0.99) | 0.98 (0.89, 1.08) | 0.96 (0.88, 1.06) | 0.95 (0.89, 1.01) |
| Under/normal classification[c] | **0.36 (0.34, 0.39)*** | **0.16 (0.15, 0.17)*** | **0.62 (0.59, 0.65)*** | **0.27 (0.25, 0.28)*** | 0.87 (0.79, 0.96) | **0.79 (0.72, 0.86)*** | **0.23 (0.21, 0.24)*** |

Odds ratios [95% confidence interval].

Models adjusted for sex, age, race, insurance type, mother comorbidity, father comorbidity, days between routine visit and lab values, and overweight/obese classification.

[a] Reference category = not meeting physical activity guidelines

[b] Reference category = not meeting screen time recommendations

[c] Reference category = overweight/obese classification

[d] n = 63,676 if visit did not contain visit for a specific cardiometabolic lab the lab was considered to be within a normal/healthy range.

* (Bolded text) indicates statistical significance with Bonferroni corrected α = 0.0036.

A1c- glycated hemoglobin; ALT—Alanine transaminase; HTN–hypertension, HDL–high density lipoprotein; LDL–low density lipoprotein

The relationship between meeting physical activity and screen time recommendations and having abnormal outcomes of modifiable cardiometabolic risk factors are displayed in Table 3. When examining only visits that included blood pressure and lab data (ie, the Lab Sample), only blood pressure was significantly associated by meeting physical activity recommendations, with meeting recommendations associated with a 9% reduced risk of having hypertension ($p = 0.0014$). When examining all visits with any missing lab data coded as normal, meeting physical activity recommendations was associated with reduced risk of having abnormal A1c by 17% ($p<0.0001$), ALT by 15% ($p<0.0001$), blood pressure by 9% ($p = 0.002$), HDL by 12% ($p = 0.0009$), and triglyceride values by 11% ($p = 0.002$; Table 3). Meeting screen time recommendations was not associated with change in risk for any of the cardiometabolic risk factors within either sample ($p = 0.027$–$0.688$).

When grouped by body weight and activity level status, similar results were observed between patients with under/normal weight regardless of if they met physical activity recommendations (in both the lab data and all data analyses; $p = 0.226$–$0.983$; Table 4). Differences in most blood pressure and lab values, with the exception of LDL, were observed between patients with under/normal weight who met physical activity recommendations and patients with overweight/obesity in both physical activity categories.

## Discussion

There is a high prevalence of children with overweight and obesity and many of them have or are at risk for developing cardiometabolic diseases [29]. Primary care pediatricians are responsible for screening for these diseases but have limited time during each visit. The use of single-

**Table 4. Odds ratios of physical activity and overweight/obese classifications with abnormal cardiometabolic risk indicators (lab and BP values).**

| | A1c | ALT | Blood Pressure | HDL | LDL | Total cholesterol | Triglycerides |
|---|---|---|---|---|---|---|---|
| **Lab sample** | | | | | | | |
| Un/N not meeting PA recommendation | 0.86 (0.50, 1.45) | 0.83 (0.61, 1.13) | 0.99 (0.91, 1.08) | 1.04 (0.84, 1.29) | 1.10 (0.73, 1.67) | 1.12 (0.78, 1.60) | 1.02 (0.81, 1.28) |
| OW/OB meeting PA recommendation | 1.28 (0.94, 1.74) | **2.13 (1.79, 2.54)**\* | **1.62 (1.55, 1.70)**\* | **2.91 (2.59, 3.26)**\* | 1.32 (1.06, 1.65) | 1.33 (1.10, 1.60) | **3.92 (3.50, 4.39)**\* |
| OW/OB not meeting PA recommendation | 1.61 (1.17, 2.23) | **2.04 (1.68,2.48)**\* | **2.00 (1.84, 2.17)**\* | **3.08 (2.67, 3.54)**\* | 1.29 (0.97, 1.73) | **1.45 (1.14, 1.85)**\* | **4.07 (3.53, 4.69)**\* |
| **Full sample**[a] | | | | | | | |
| Un/N not meeting PA recommendation | 0.96 (0.81, 1.14) | 1.0 (0.86, 1.16) | 0.99 (0.91, 1.08) | 0.99 (0.86, 1.15) | 1.01 (0.84, 1.21) | 1.02 (0.85, 1.21) | 1.00 (0.87, 1.16) |
| OW/OB meeting PA recommendation | **2.59 (2.39, 2.81)**\* | **5.97 (5.58, 6.39)**\* | **1.56 (1.49, 1.64)**\* | **3.63 (3.39, 3.89)**\* | 1.15 (1.04, 1.27) | **1.26 (1.14, 1.39)**\* | **4.31 (4.03, 4.60)**\* |
| OW/OB not meeting PA recommendation | **3.68 (3.27, 4.13)**\* | **7.28 (6.63, 7.99)**\* | **1.91 (1.76, 2.07)**\* | **4.35 (3.95, 4.79)**\* | 1.17 (0.98, 1.41) | **1.40 (1.18, 1.65)**\* | **5.04 (4.58, 5.55)**\* |

Odds Ratio (95% Confidence Interval).

Reference category: Under/normal weight meeting physical activity recommendation.

Models adjusted for meeting screen time recommendation, sex, age, race, insurance type, mother comorbidity, father comorbidity, and days between routine visit and lab values.

[a] n = 63,676 if visit did not contain visit for a specific lab the lab was considered to be within a normal/healthy range.

\* (Bolded text) indicates statistical significance with Bonferroni corrected α = 0.0036.

UN/N–under/normal weight body mass index category; OW/OB–overweight/obese body mass index category; PA–physical activity; A1c- glycated hemoglobin; ALT— Alanine transaminase; BP–blood pressure, HDL–high density lipoprotein; LDL–low density lipoprotein.

item screeners for lifestyle behaviors has the potential to be a tool to quickly identify unhealthy behaviors and alert clinicians to provide tailored counseling on specific behavioral risk factors. The primary findings from this study indicate youth who met physical activity recommendations, as collected during well child visits using a single-item screener, had lower risk for modifiable cardiometabolic risk factors. In addition, youth with overweight/obesity who met physical activity recommendations had lower risk of modifiable cardiometabolic risk factors compared to youth with overweight/obesity not meeting recommendations. These data further demonstrate the benefit of including lifestyle assessment questions in pediatric primary care to provide more context and identify children who may be at a higher risk for adverse cardiometabolic health, regardless of weight status.

Meeting physical activity recommendations was associated with lower cardiometabolic risk factors, with the exceptions of LDL and total cholesterol. Meeting physical activity recommendations was associated with better cardiometabolic risk factor outcomes between 9–17%. Similar results are observed in children, children with overweight and obesity with exercise having a small but significant effect on ALT, blood pressure, HDL, and triglycerides [30, 31]. Risk factors, such as LDL and total cholesterol, are influenced more by diet than activity, and it is therefore not surprising that meeting activity guidelines had limited association with healthier outcomes [32]. It should be noted that there are many health behaviors, in addition to physical activity and screen time, that are related to cardiometabolic risk (e.g., diet and sleep). Additionally, many of these health behaviors can be interrelated (e.g., screen time is associated with snacking / poor dietary habits) [33, 34] and interventions targeting multiple health behaviors may observe enhanced benefits for cardiometabolic health.

Our results indicate that weight classification had a strong impact on modifiable cardiometabolic risk factors. Previous research in youth support similar results when comparing the

impact of weight status and fitness on cardiometabolic risk factors [35–37]. It is promising that results support that youth with overweight/obesity who were active can still experience cardiometabolic risk factor benefits compared to their counterparts who are not meeting physical activity recommendations. Our findings observed comparable results regardless of meeting physical activity recommendations in youth within the under/normal weight classification. These results are similar to previous research that observed no difference based on fitness level of children in this weight category on cardiometabolic risk scores [35]. The present study furthers the literature by examining physical activity, rather than fitness, and by using a measure that is readily available to clinicians. Electronic medical records are an easy way to assess healthy lifestyle behaviors and counsel children with overweight/obesity who may be at a higher risk for cardiometabolic health conditions and who are not active [30].

Contrary to expectations, meeting screentime recommendations did not have an impact on any modifiable cardiometabolic risk factors. Previous literature provides strong reports between meeting screen time recommendations and weight status and waist circumference, but research extending to other cardiometabolic risk factors report more varied results [16, 38–41]. The lack of statistical significance in the present study may be due to the lifestyle assessment only capturing a portion of behaviors (eg, responses may not consider all types of screens [phones, iPads, etc]), or the dichotomized nature of analyses (ie, meeting or not meeting recommendations). Another potential reason for the lack of differences observed from meeting recommendations may be the low amount of screen time considered for meeting recommendations (ie, 2 hours).

## Clinical implications

A unique aspect of our study is the use of data collected from EMRs. There is growing support for the integration of brief lifestyle assessment questions (eg, physical activity, screen time, diet) into EMRs. As demonstrated by the current study and previous literature, responses to these types of questions may indicate patients who are at a higher risk of adverse cardiometabolic health [16, 17, 42]. Viewing patient responses to lifestyle assessments will enable clinicians to develop and provide appropriate recommendations and/or counseling material for individuals who could improve their lifestyle behaviors and thus improve their modifiable cardiometabolic risk factors [19, 20, 43]. For example, when a child has elevated values on one or more cardiometabolic risk factors, the provider can review their lifestyle behaviors to determine which may warrant improvement and discuss options with the child and their family.

## Strengths and limitations

A key strength of this study is it was conducted on a large dataset within a health system that serves a large number of Black and Latino children. This is a strength as the racial diversity of the sample makes the findings more generalizable compared to studies examining this within a sample of a single race. Additionally, it examines relationships between lifestyle behavior assessments and modifiable cardiometabolic risk factor lab values that are already implemented into a healthcare system, allowing results to be immediately viewable to clinicians. Limitations within this study include that the lifestyle behaviors were reported, rather than objectively measured, and therefore were subject to bias, including social desirability bias. Dichotomizing physical activity and screentime behaviors into meeting or not meeting recommendations oversimplifies the complexity of these behaviors which potentially overlooks the nuance of each behavior and does not evaluate health benefits associated at different intervals of engagement in the behaviors. Additionally, other lifestyle behaviors (ie, diet) are impactful these cardiometabolic health factors and/or may have moderating effects on the outcomes of

interest but were not able to be included in the analysis. While the dataset is large and diverse, there are limitations to generalizability of the results including that the data is only from one health system that serves a specific population (primarily un- and under-insured lower-income families) and children with type 1 diabetes were excluded from the study. For the lab values themselves, an inherent limitation within the healthcare system is that fasting values cannot be confirmed (eg, triglycerides) and LDL values were calculated, rather than measured. Blood pressure and lab values are proxies (albeit well-recognized ones) for cardiometabolic health, and there are reasons other than risk for cardiometabolic disease that BP and these lab values can be elevated. When examining parent comorbidities/health conditions, data were insufficient on some conditions which may contribute to child health risk, including nonalcoholic fatty liver disease, myocardial infarction, stroke, cerebrovascular accident, obstructive sleep apnea, or polycystic ovarian syndrome. A final limitation inherent with the cross-sectional nature of the data is the inability to assess causation, or if the lifestyle behavior or abnormal cardiometabolic risk factor appeared first.

## Conclusion

Healthcare professionals have a unique opportunity to impact health and provide counseling on lifestyle behaviors. Results from this study support the benefits of physical activity on cardiometabolic health in children. Further, single-time questions within EMRs can support clinicians in identifying youth at risk for adverse modifiable cardiometabolic risk factors associated with not meeting recommendations.

## Supporting information

**S1 Checklist. Human participants research checklist.**
(DOCX)

**S1 Table. Guidelines used for normal and abnormal blood pressure and cardiometabolic lab values.** A1c- glycated hemoglobin; ALT—Alanine transaminase; HDL–high density lipoprotein; LDL–low density lipoprotein.
(PDF)

**S1 File.**
(PDF)

**S2 File.**
(DOCX)

## Acknowledgments

The authors thank Gary Kruger for pulling the data and Keith Feldman for aiding in merging data.

## Author Contributions

**Conceptualization:** Bethany Forseth, Sarah Hampl, Jordan A. Carlson, Ann Davis.

**Data curation:** Bethany Forseth, Kelsee Halpin, Tarin Phillips.

**Formal analysis:** Bethany Forseth, Janelle R. Noel-MacDonnell, Jordan A. Carlson.

**Investigation:** Bethany Forseth, Sarah Hampl, Kelsee Halpin.

**Methodology:** Bethany Forseth, Janelle R. Noel-MacDonnell, Jordan A. Carlson, Robin P. Shook.

**Project administration:** Sarah Hampl, Robin P. Shook.

**Resources:** Sarah Hampl, Ann Davis.

**Software:** Janelle R. Noel-MacDonnell.

**Supervision:** Janelle R. Noel-MacDonnell, Sarah Hampl, Jordan A. Carlson, Kelsee Halpin, Ann Davis, Robin P. Shook.

**Writing – original draft:** Bethany Forseth, Jordan A. Carlson.

**Writing – review & editing:** Bethany Forseth, Janelle R. Noel-MacDonnell, Sarah Hampl, Jordan A. Carlson, Kelsee Halpin, Ann Davis, Tarin Phillips, Robin P. Shook.

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
