## [Decision Letter · Decision Letter 0]

1 Mar 2024

PONE-D-23-36566Relationship between youth cardiometabolic health and physical activity in medical recordsPLOS ONE

Dear Dr. Forseth,

Thank you for submitting your manuscript to PLOS ONE. After careful consideration, we feel that it has merit but does not fully meet PLOS ONE’s publication criteria as it currently stands. Therefore, we invite you to submit a revised version of the manuscript that addresses the points raised during the review process.

 The reviewers have identified aspects of the abstract that require revision, and have identified limitations of this study that require acknowledgement and/or discussion. In addition, please clarify the structure of the mixed models used (lines 177-178), particularly what was considered as random vs. fixed effects.    ** **==============================

We look forward to receiving your revised manuscript.

Kind regards,

Toby Edward Mansell, PhD, MBiostat

Academic Editor

PLOS ONE

Journal Requirements:

2. In the online submission form, you indicated that data cannot be shared publicly because of it is identifiable patient data from electronic medical records. Data may be requested from the Children's Mercy Hospital Institutional Data Access / Ethics Committee (contact bhanson4@kumc.edu for more details and guidance) for researchers who meet the criteria for access to confidential data.

Reviewers' comments:

Reviewer's Responses to Questions

**Comments to the Author**

1. Is the manuscript technically sound, and do the data support the conclusions?

Reviewer #1: Yes

Reviewer #2: Partly

2. Has the statistical analysis been performed appropriately and rigorously? 

Reviewer #1: Yes

Reviewer #2: Yes

3. Have the authors made all data underlying the findings in their manuscript fully available?

Reviewer #1: Yes

Reviewer #2: Yes

4. Is the manuscript presented in an intelligible fashion and written in standard English?

Reviewer #1: No

Reviewer #2: Yes

5. Review Comments to the Author

Reviewer #1: The study contributes valuable information to the field and highlights the potential benefits of incorporating lifestyle assessment questions into electronic medical records for identifying children at higher risk for adverse cardiometabolic health.

The Author is aware about the study limitations but the following could be added:

Dichotomizing physical activity and screen time into meeting or not meeting recommendations oversimplifies the complexity of physical activity and screen time behaviors, potentially overlooking nuanced variations in lifestyle.

Although the study used a large data set with racial diversity, its generalizability is limited due to many factors, such as the exclusion of missed data and type I diabetics.

Self-reported data can be subject to biases, including social desirability bias, where respondents may provide answers they believe are socially acceptable rather than accurate

Reviewer #2: Dear Editor,

I would like to express my appreciation for having the opportunity to review the manuscript titled: "Relationship between youth cardiometabolic health and physical activity in medical records." The manuscript is well-written, and the author has put in a commendable effort to prepare it. However, there are some comments that I believe need to be addressed:

Abstract:

-The background section lacks clarity and should be rewritten to present the problem instead of just reporting results.

-Abbreviations must be mentioned in the abstract or before the start of the manuscript text in the metadata.

-The conclusion does not effectively summarize the findings and should be rewritten.

The authors did not evaluate the dietary intake and diet habits which is very important and make a very serious limitation for their study.

In a cross-sectional study investigating the correlation between physical activity and cardiometabolic risk factors, evaluating and assessing dietary intake and habits is crucial for several reasons:

Understanding Confounding Variables: Dietary intake and habits can act as confounding variables in the relationship between physical activity and cardiometabolic risk factors. Without controlling for diet, it's challenging to ascertain whether observed effects are truly due to physical activity or if they are influenced by dietary factors.

Assessing Lifestyle Patterns: Diet and physical activity are intertwined aspects of lifestyle. Evaluating both allows researchers to understand the holistic lifestyle patterns of participants. This comprehensive approach helps in identifying potential interactions and associations between diet, physical activity, and cardiometabolic risk factors.

Identifying Mediating Effects: Dietary factors may mediate the relationship between physical activity and cardiometabolic risk factors. For instance, certain dietary patterns might amplify or mitigate the effects of physical activity on cardiometabolic health. By assessing diet, researchers can explore these potential mediating mechanisms.

Enhancing Generalizability: Including dietary assessment enhances the generalizability of findings. Diet varies across populations and regions due to cultural, economic, and environmental factors. Assessing dietary intake and habits allows researchers to account for these differences and generalize findings to broader populations.

Informing Intervention Strategies: Understanding participants' dietary habits provides valuable insights for designing effective intervention strategies. For instance, if certain dietary patterns exacerbate cardiometabolic risk factors, interventions can target dietary modifications alongside promoting physical activity.

Comprehensive Risk Assessment: Cardiometabolic risk is influenced by multiple factors, including both physical activity and diet. Assessing dietary intake alongside physical activity provides a more comprehensive understanding of individuals' overall risk profiles. This holistic approach enables better risk stratification and personalized interventions.

Data Interpretation and Adjustment: When analyzing data, researchers can adjust for dietary variables to isolate the specific effects of physical activity on cardiometabolic risk factors. This adjustment helps in elucidating the independent contribution of physical activity while accounting for potential dietary confounders.

Promoting Public Health Recommendations: Findings from studies examining the relationship between physical activity, diet, and cardiometabolic risk factors can inform public health recommendations. Understanding how dietary factors interact with physical activity informs guidelines for promoting healthier lifestyles and reducing cardiometabolic risk at the population level.

6. PLOS authors have the option to publish the peer review history of their article (what does this mean?). If published, this will include your full peer review and any attached files.

Reviewer #1: **Yes: **Dr. Anees Alyafei

Reviewer #2: No

---

## [Author Response · Author response to Decision Letter 0]

12 Apr 2024

Dear Editor Mansell, Editorial Board, and reviewers, 

Thank you for your review of our manuscript, ‘Relationship between youth cardiometabolic health and physical activity in medical records.’ We appreciate the time and thoughtfulness that you put into the reviews. We have edited the paper based on your comments and have also directly responded to the comments below. We apologize in advance, track changes were not on when we revised the abstract, so changes to not appear in the ‘track changes’ version, but this was updated!

Journal requirements.

Thank you for the helpful links! We have updated headers, figure / table /supplemental file names, references, etc. 

2. In the online submission form, you indicated that data cannot be shared publicly because of it is identifiable patient data from electronic medical records. Data may be requested from the Children's Mercy Hospital Institutional Data Access / Ethics Committee (contact bhanson4@kumc.edu for more details and guidance) for researchers who meet the criteria for access to confidential data.

Thank you for supporting transparency with data and publicly available data. Our dataset was created from electronic medical records from one hospital (the hospital is identified by name in the paper on the ethics approval line) and contains many data aspects that may compromise patient privacy. We did reach out to our IRB to identify best options for sharing the data. The IRB responded that our study was not approved to share patient data. We are able to share deidentified data with other researchers who complete a data user sharing agreement with Children’s Mercy; this agreement will need to be completed with the Children’s Mercy Office of Research Business Partners. As corresponding author, if another researchers requests the data I am happy to help with working through this process the data user agreement with the Children’s Mercy Institution. 

3. your ethics statement should only appear in the Methods section of your manuscript. If your ethics statement is written in any section besides the Methods, please move it to the Methods section and delete it from any other section. Please ensure that your ethics statement is included in your manuscript, as the ethics statement entered into the online submission form will not be published alongside your manuscript.

We have included an ethic statement in the methods section (line 112) and nowhere else in the manuscript. Per this request, we have removed the ‘ethics approval’ statement from the end of the main manuscript document following the conclusion. 

4. Please include captions for your Supporting Information files at the end of your manuscript, and update any in-text citations to match accordingly. 

We have included a caption for our supporting information table that matches the text. 

We have updated our supporting information file (supplemental table 1) and modified our IRB documents to ‘other’ files. 

Editor 

Please clarify the structure of the mixed models used (lines 177-178), particularly what was considered as random vs. fixed effects.

Thank you for catching this missing information. We have added this information into the manuscript (Lines 178-180 – in the track changes document). 

Reviewer 1 

The study contributes valuable information to the field and highlights the potential benefits of incorporating lifestyle assessment questions into electronic medical records for identifying children at higher risk for adverse cardiometabolic health.

The Author is aware about the study limitations but the following could be added:

Dichotomizing physical activity and screen time into meeting or not meeting recommendations oversimplifies the complexity of physical activity and screen time behaviors, potentially overlooking nuanced variations in lifestyle.

This is an excellent point. We have included a comment on this into the limitation (lines 302-305– in the track changes document). 

Although the study used a large data set with racial diversity, its generalizability is limited due to many factors, such as the exclusion of missed data and type I diabetics.

This has been added to the limitations on generalizability the population (lines 307 – 310– in the track changes document). 

Self-reported data can be subject to biases, including social desirability bias, where respondents may provide answers they believe are socially acceptable rather than accurate

This is correct, thank you for pointing out this specific bias. We have included this into the limitations (line 302– in the track changes document). 

Reviewer 2

I would like to express my appreciation for having the opportunity to review the manuscript titled: "Relationship between youth cardiometabolic health and physical activity in medical records." The manuscript is well-written, and the author has put in a commendable effort to prepare it. However, there are some comments that I believe need to be addressed:

Abstract:

-The background section lacks clarity and should be rewritten to present the problem instead of just reporting results.

Thank you for this comment, we have added additional information into the background to provide reader with more formation on the lack of research on this topic (lines 48-50– in the track changes document). We apologize – track changes were not tracked in the abstract. 

-Abbreviations must be mentioned in the abstract or before the start of the manuscript text in the metadata.

Thank you for catching this. We have now defined ‘Odds Ratio’ (OR) for the statistical

results within the abstract (line 61– in the track changes document). We apologize – track changes were not tracked in the abstract. 

-The conclusion does not effectively summarize the findings and should be rewritten.

We have re-worked the conclusion to better represent our findings (Line 65– in the track changes document). We apologize – track changes were not tracked in the abstract. 

The authors did not evaluate the dietary intake and diet habits which is very important and make a very serious limitation for their study.

In a cross-sectional study investigating the correlation between physical activity and cardiometabolic risk factors, evaluating and assessing dietary intake and habits is crucial for several reasons:

Understanding Confounding Variables: Dietary intake and habits can act as confounding variables in the relationship between physical activity and cardiometabolic risk factors. Without controlling for diet, it's challenging to ascertain whether observed effects are truly due to physical activity or if they are influenced by dietary factors.

Assessing Lifestyle Patterns: Diet and physical activity are intertwined aspects of lifestyle. Evaluating both allows researchers to understand the holistic lifestyle patterns of participants. This comprehensive approach helps in identifying potential interactions and associations between diet, physical activity, and cardiometabolic risk factors.

Identifying Mediating Effects: Dietary factors may mediate the relationship between physical activity and cardiometabolic risk factors. For instance, certain dietary patterns might amplify or mitigate the effects of physical activity on cardiometabolic health. By assessing diet, researchers can explore these potential mediating mechanisms.

Enhancing Generalizability: Including dietary assessment enhances the generalizability of findings. Diet varies across populations and regions due to cultural, economic, and environmental factors. Assessing dietary intake and habits allows researchers to account for these differences and generalize findings to broader populations.

Informing Intervention Strategies: Understanding participants' dietary habits provides valuable insights for designing effective intervention strategies. For instance, if certain dietary patterns exacerbate cardiometabolic risk factors, interventions can target dietary modifications alongside promoting physical activity.

Comprehensive Risk Assessment: Cardiometabolic risk is influenced by multiple factors, including both physical activity and diet. Assessing dietary intake alongside physical activity provides a more comprehensive understanding of individuals' overall risk profiles. This holistic approach enables better risk stratification and personalized interventions.

Data Interpretation and Adjustment: When analyzing data, researchers can adjust for dietary variables to isolate the specific effects of physical activity on cardiometabolic risk factors. This adjustment helps in elucidating the independent contribution of physical activity while accounting for potential dietary confounders.

Promoting Public Health Recommendations: Findings from studies examining the relationship between physical activity, diet, and cardiometabolic risk factors can inform public health recommendations. Understanding how dietary factors interact with physical activity informs guidelines for promoting healthier lifestyles and reducing cardiometabolic risk at the population level.

Thank you for you recommendation of the importance of dietary measures and their inclusion within health behavior research. We agree dietary measures are important components of health and how their addition could strengthen our paper and the field of research. This is definitely something we hope to consider for future papers, but due to limitations of the data and the scope of this paper, we are unable to incorporate diet behaviors/intake into this specific manuscript. We have recognized the importance of diet and other health behaviors related to cardiometabolic health into the discussion (lines 259-265) and highlight the scientific limitations (data interpretation and moderating effects) of this paper without the ability to report these variables (lines 305-307– in the track changes document).

---

## [Editor Report · Decision Letter 1]

29 Apr 2024

Relationship between youth cardiometabolic health and physical activity in medical records

PONE-D-23-36566R1

Dear Dr. Forseth,

We’re pleased to inform you that your manuscript has been judged scientifically suitable for publication and will be formally accepted for publication once it meets all outstanding technical requirements.

Kind regards,

Toby Edward Mansell, PhD, MBiostat

Academic Editor

PLOS ONE

---

## [Editor Report · Acceptance letter]

10 May 2024

PONE-D-23-36566R1 

PLOS ONE

Dear Dr. Forseth, 

I'm pleased to inform you that your manuscript has been deemed suitable for publication in PLOS ONE. Congratulations! Your manuscript is now being handed over to our production team.

Kind regards, 

on behalf of

Dr. Toby Edward Mansell 

Academic Editor

PLOS ONE